# News Insights into the Host-Parasite Interactions of Amyloodiniosis in European Sea Bass: A Multi-Modal Approach

**DOI:** 10.3390/pathogens11010062

**Published:** 2022-01-04

**Authors:** Michela Massimo, Donatella Volpatti, Marco Galeotti, James E. Bron, Paola Beraldo

**Affiliations:** 1Section of Animal and Veterinary Sciences, Department of Agricultural, Food, Environmental and Animal Sciences, University of Udine, 33100 Udine, Italy; michela.massimo@uniud.it (M.M.); donatella.volpatti@uniud.it (D.V.); marco.galeotti@uniud.it (M.G.); 2Institute of Aquaculture, University of Stirling, Stirling FK9 4LA, UK; j.e.bron@stir.ac.uk

**Keywords:** dinoflagellate, *Amyloodinium ocellatum*, histology, immunohistochemistry, confocal laser scanning microscopy, aquaculture, European sea bass

## Abstract

Amyloodiniosis is a disease resulting from infestation by the ectoparasitic dinoflagellate *Amyloodinium ocellatum* (AO) and is a threat for fish species such as European sea bass (ESB, *Dicentrarchus labrax*), which are farmed in lagoon and land-based rearing sites. During the summer, when temperatures are highest, mortality rates can reach 100%, with serious impacts for the aquaculture industry. As no effective licensed therapies currently exist, this study was undertaken to improve knowledge of the biology of AO and of the host-parasite relationship between the protozoan and ESB, in order to formulate better prophylactic/therapeutic treatments targeting AO. To achieve this, a multi-modal study was performed involving a broad range of analytical modalities, including conventional histology (HIS), immunohistochemistry (IHC) and confocal laser scanning microscopy (CLSM). Gills and the oro-pharyngeal cavity were the primary sites of amyloodiniosis, with hyperplasia and cell degeneration more evident in severe infestations (HIS). Plasmacells and macrophages were localised by IHC and correlated with the parasite burden in a time-course experimental challenge. CLSM allowed reconstruction of the 3D morphology of infecting trophonts and suggested a protein composition for its anchoring and feeding structures. These findings provide a potential starting point for the development of new prophylactic/therapeutic controls.

## 1. Introduction

*Amyloodinium ocellatum* (Brown and Hovasse, 1946) (Thoracosphaeraceae, Thoracosphaerales, Dinophyceae, Chromista) is an ectoparasitic dinoflagellate distributed in marine and brackish-water environments in both tropical and temperate regions worldwide. The sessile trophont is the parasitic stage and feeds directly from host cells, causing gill and skin infestation known as amyloodiniosis. *A. ocellatum* (AO) trophont pathogenicity results from attachment to host epithelial cells through root-like rhizoids and a mobile stomopode, with the parasite damaging and killing host cells [1]. Consequently, pathology depends on parasite load and is mainly proportional to the extent of gills infestation. Amyloodiniosis symptoms, more evident in severe or advanced infestations, include anorexia, dyspnoea, jerky swimming and gasping. Mortality can occur as a consequence of osmoregulatory and respiratory impairment and, in some cases, secondary bacterial infection. AO represents a serious problem for several cultured fish species (*Chanos chanos* (Forsskål, 1775); *Diplodus puntazzo* (Walbaum, 1792); *Rachycentron canadum* (Linnaeus, 1766); *Scophtalmus maximus* (Linnaeus, 1758); *Solea solea* (Linnaeus, 1758) and *Sparus aurata* (Linnaeus, 1758)) and amyloodiniosis can kill the host in less than 12 h, with acute mortality around 100%, depending on the farming conditions, parasite burden, fish species and season [2,3,4,5,6]. Generally, the infestation affects land- or lagoon-based rearing sites, characterized by shallow water and poor water exchange/recirculation, which together provide favourable conditions for AO proliferation. High temperatures also encourage optimal parasite development, and in the warmest months, *A. ocellatum* is capable of causing high mortality rates and significant economic losses. To date, there are no efficacious prophylactic or therapeutic controls for amyloodiniosis.

In order to develop more effective prophylactic/therapeutic controls, a better understanding of the mechanisms underlying host-parasite interactions is a key requirement. Detailed descriptions of changes provoked by AO in the gill epithelium of different host species have been provided by previous authors, who furnished consistent reports of these lesions alluding also to the mechanisms involved in the host response [4,5,7,8,9,10,11]. Ultrastructural studies have better clarified the morphology of the parasite and the stomopode conformation/composition, including aspects of parasite-host interaction with three cyprinodontid species [12]. Scanning electron microscopy was used to detail local erosion and the distortion of the gill epithelium at the site of trophont attachment on naturally infested fish living in the Salton Sea Lake [2,13].

Knowledge concerning European sea bass (ESB, *Dicentrarchus labrax* (Linnaeus, 1758) infestation by AO is limited [14,15]. The specific anti-AO antibody response of hosts, measured by ELISA and the host pro-inflammatory response, measured through quantification of marker transcripts, have been investigated [10,16]. The level of expression of molecules synthesized as part of innate and adaptive immunity during a natural infestation and the localization of leukocyte populations recruited to cope with the outbreak have also been studied recently [17]. Knowledge concerning the immunohistochemistry/detailed histopathology of infestations is still lacking and for this study it was considered that a confocal microscopy approach would prove productive as there have been no previous confocal studies conducted on infestation of ESB or any other fish species infested by the dinoflagellate.

Hence, the aim of the present study was to deepen our understanding of the mechanisms underlying the host-parasite relationship between *A. ocellatum* and the European sea bass, which is a valuable species for Mediterranean aquaculture as well as a documented host of this parasite. To achieve this, a multi-modal study has been performed involving histology (HIS), immunohistochemistry (IHC), and confocal laser scanning microscopy (CLSM).

## 2. Results

### 2.1. Histology (HIS)

All samples of ESB gills and heads, coming from naturally and experimentally infested fish, contributed to the histological description reported below. Gills and the entire oro-pharyngeal cavity were the primary sites of infestation in ESB (Figure 1a,b,h) and, albeit in some surveyed infestations the skin was also parasitized, it did not display the dusty appearance observed in other fish species subject to high infestations. AO infested ESB died without obvious gross skin lesions, as confirmed by histological observations.

In the early stages of, or in slight infestations (less than ten trophonts/primary lamella) no histologically observable lesions were detectable, although a mild epithelial hyperplasia was occasionally observed around the single trophont adhesion site, associated with epithelial cell degeneration (Figure 1e).

The parasite load was considered very high when more than fifty trophonts were present on a singular primary lamella. In the first 36–48 h of heavy infestation, severe and diffuse epithelial degeneration in the oro-pharyngeal cavity was present, characterized by hydropic degeneration of gill epithelia and chloride cells, oedema and necrosis (Figure 1c,e,g); subsequently it was possible to observe hyperplasia of the gill epithelium, especially in the distal third of the primary lamellae (Figure 1c).

The same histological pattern was observed in the pseudobranchs (Figure 1f), while marked epithelial hyperplasia was the primary pathological change observed in the buccal cavity (Figure 1b), pharynx (Figure 1h) and gill arch epithelium.

Trophonts were also well distributed in the nasal cavities, with diffuse epithelial degeneration and cellular infiltration being observed. Marked gill epithelial hyperplasia induced a pattern in which synechiae were easily visible at lamellar tips (Figure 1c). In the gill lamellae the inflammatory cells were difficult to differentiate, however along the filaments, lymphocytes, macrophages, and mast cells were discernible (Figure 1d). The same cell infiltrate pattern was more evident in the buccal cavity and pharynx, where mucous cells were very abundant in the epithelium (Figure 1h). Moreover, a small number of rodlet cells was often present along the lamellar epithelium, most of all at the base of secondary lamellae. *A. ocellatum* trophonts feed and anchor on multiple epithelial cells simultaneously, inducing extensive damage to the vascular system as well as provoking the rupture of pillar cells and formation of lamellar aneurysms or micro haemorrhages (Figure 1g). These changes were observed at the necropsy in the form of anaemia in gills.

Histologically, AO causes mainly a gill inflammatory pattern that can be framed as hyperplastic parasitic branchitis, although the hyperplasia extent depends on the course of infestation.

### 2.2. Immunohistochemistry (IHC)

The following description concerns the general IHC outcomes in specimens collected from the experimental infestation trial.

The hyperplastic gill lamellae revealed the presence of cell populations showing positivity for antibodies specifically labelling ESB IgM and iNOS. Control sections of gills collected from infected ESB, in which the primary antibody was replaced by dilution buffer, were always negative (Figure 2a,e). The polyclonal antibody specific for ESB IgM marked cells whose morphology corresponded to that of plasma cells and in some cases to Ig-bearing macrophages. They were occasionally detectable in the epithelium of branchial lamellae or at the base of primary lamellae. The number of positive cells progressively increased throughout the post infestation phase (Figure 2b), becoming relevant at 30 days (Figure 2c,d).

Macrophage-like cells positive for iNOS were mainly localized in the gill areas where secondary lamellae were hyperplastic and fused (Figure 2f). Furthermore, iNOS-positive labelling was also associated with antigens expressed by trophonts in adhesion (Figure 2f). The application of the iNOS marker to gill tissues collected from healthy ESB showed detection of very few positive cells, whose number was negligible when compared to positive labelling in the pathological samples.

### 2.3. Confocal Laser Scanning Microscopy (CLSM)

Under confocal microscope eosin exhibits maximum absorption and emission at 527 and 550 nm respectively, with both peaks lying in the green light range. Four µm haematoxylin and eosin (H-E) slides from our research group archives in Udine (UNIUD) provided good results even if the thickness of the sections (4 μm) was not ideal for confocal reconstruction of the full trophont depth. The images captured from these samples gave the opportunity to better visualize the protein granules contained in the cytoplasm of trophonts, and for some parasites, also the conformation of the stomopode (Figure 3a) and rhizoid disposition. On the other hand, the extent of the parasite wall was not clearly defined.

Further, 4,6-diamidino-2-phenylindole,dihydrochloride (DAPI) was used, with peak excitation wavelengths at 360 nm, and blue fluorescence (460 nm). In the samples mounted with DAPI mounting medium it was possible to discriminate the nuclear details in both gill epithelial cells and trophonts (Figure 3b). In these preparations, the trophont wall was not clearly defined as well as the stomopode and rhizoid details.

TRITC-phalloidin is excited at 544 nm and emits at 572 nm, producing an orange-red fluorescence. Slides mounted with TRITC-phalloidin did not provide a selective staining, probably due to the overlapping of the fluorescence emission with that of eosin.

Calcofluor white (CFW) binds to β1–3 and β1–4 polysaccharides such as those found in cellulose. The absorption spectrum for aqueous CFW solution peaks at 347 nm and, when excited with UV radiation, fluoresces with an intense blue colour. Propidium iodide (PI), a nuclear stain, presents an excitation wavelength at 485 nm, and fluorescence with wavelengths greater than 595 nm, providing a red wavelength emission. In the samples processed by using the CFW and PI double staining, nuclear elements of the host gill cells appeared red stained in contrast with the green background. In the parasites, a pale reddish colour was visible in the nuclear region even if covered by the eosin green emission of the protein granules. CFW specifically bound to the cellulose wall staining it in blue (Figure 3c).

The lectins used for this investigation were conjugated to fluorophores that provided maximum fluorescence and optimal staining characteristics. In order to avoid fluorescence fading, slides labelled with fluorescent lectins were visualized on the same day. Wheat germ agglutinin (WGA) contained in the fluorescein labelled Lectin Kit I shows an excitation maximum at 495 nm and an emission maximum at 515 nm. Images captured using this lectin provided good results in terms of labelling specificity. WGA bound only to the cellulose wall of AO trophonts, marking its thickness and delineating its shape (Figure 3d). In this case, the trophont visualized was quite spherical suggesting it was in the process of transforming to a tomont. The other cellular components of both parasite and host gill epithelium presented no fluorescent labelling.

WGA plus rhodamine lectin showed slightly different fluorescent parameters compared to the previous lectin used, with a maximum excitation at 550 nm and an emission maximum at 575 nm. In addition, this protein labelled the cellulose wall of trophonts (Figure 3e). However, in this case it was also possible to observe the conformation of the armoured theca, which displayed the typical pear shape of trophonts. No labelling was detected in the negative controls (slides incubated with the dilution buffer only, Figure 3f).

## 3. Discussion

The combination of approaches employed here has provided a clearer understanding of the interactions and impacts of the protozoan *Amyloodinium ocellatum* on European sea bass juveniles and adults and has also allowed clarification of aspects of the immune response mechanisms adopted by fish to overcome the infestation.

The histological results reveal the pathological alterations that characterize heavy infestations in European sea bass, regardless of fish size. The parasitic branchitis is the main inflammatory pattern observed in severe amyloodiniosis, gills and the entire oro-pharyngeal cavity, nostrils included, being the primary sites of infestation in ESB. The skin, even if abundantly infested by trophonts, did not show clear lesions nor the typical dusty appearance as described in other species, and from which derives the common name of the disease (marine velvet disease). Ultimately, the observed histological findings are consistent with those reported in previous studies conducted on other fish species naturally infested by *A. ocellatum* [2,4,5,7,8,13,18] and in *A. ocellatum* infested ESB [10]. In general, the inflammatory cell infiltrates were scarce in the gills, in agreement with the description provided by Paperna [7] on gilthead sea bream (*Sparus aurata*).

The IHC data reported here contribute to knowledge of morphological and functional aspects of the gill’s reactivity to AO. For this study, the two antibody markers employed were selected according to their relevance as indicators of tissue reactivity in terms of innate/adaptive immune response and on their previously assessed reactivity on ESB tissues by the authors.

The European sea bass polyclonal anti-IgM antibody specifically labelled plasma cells and macrophages bearing immunoglobulins, with the distribution of these cells varying across the time-course of the infestation. In the present study, the use of this antibody provided further confirmation of the ability of fish to develop specific immunity against *A. ocellatum* [16,19,20,21,22,23] and allowed the localization and enumeration of positive cells, correlated with the progression of the infestation.

iNOS antibody was newly investigated in this research following the previous results reported in Byadgi et al. [17]. The inducible nitric oxide synthase (iNOS) enzyme, which possesses both signal and bactericidal functions, was used as a measure of the innate response of ESB to AO infestation. According to our previous findings, the positivity was localized mainly in the secondary lamella epithelium, while the number of macrophage cells increased proportionally to the parasite burden. In the present investigation the labelling was again observed in the cytoplasm of trophonts. As already mentioned [17], this positivity could be explained as reflecting a product of the protozoan, with iNOS being a phylogenetically conserved molecule [24]. On the other hand, the positivity could derive from host material absorbed by AO during feeding [12]. This latter aspect was further explored in a previous study by means of in situ hybridization, with the use of a probe specific for the Chemokine CC1 transcript highlighting the presence of the mRNA sequence of this chemotactic cytokine inside the cytoplasm of trophonts, thus confirming the phagotrophic behaviour of the protozoan [25].

The IHC approach not only improved understanding of the immune response mechanisms adopted by European sea bass over the time-course of an experimental infestation, but also confirmed the activation of GIALT (gill associated lymphoid tissue) components such as phagocytic as well as the antibody producing cells.

Confocal laser scanning microscopy (CLSM) on *A. ocellatum* trophonts anchored to the gill epithelium of ESB has been applied in this study for the first time. The CLSM investigations were performed in order to develop staining protocols aimed at better detailing the anatomy of AO parasitic stage through an alternative approach. The purpose was to capture images of the stomopode and rhizoids to elucidate the way of interaction with host epithelium and to comprehend their composition. Through CLSM it was possible to capture anatomical particulars of the parasite and to obtain 3D images of AO.

Eosin is a known xanthene dye, whose high-fluorescence emission [26] lies in the green light range. In this research, eosin was used as counterstain in all the slides examined and, probably because of the overlapping of the fluorescence emission, tetramethylrhodamine (TRITC)–phalloidin did not provide any specific labelling of the cytoskeleton. In fact, the bicyclic heptapeptide phalloidin should specifically bind to F-actin filament ends [27].

Originally synthesized for trypanosomiasis treatment, 4,6-diamidino-2-phenylindole,dihydrochloride (DAPI) was revealed to be a specific DNA fluorescent probe binding strongly to adenine-thymine rich regions [28], and producing a blue fluorescence when excited by UV light. In addition, propidium iodide (PI) was used in this study to label the nuclear details, being an intercalating agent that binds to DNA or RNA, causing orange fluorescence [29].

Calcofluor white M2R (CFW) or fluorescent brightener 28 is one of a group of compounds known as fluorescent brighteners or optical brighteners, or “whitening agents” as it is used to whiten and to prevent yellowing of papers and fabrics [30]. CFW is a non-specific fluorochrome that binds to β1–3 and β1–4 polysaccharides normally found in chitin and cellulose, contained in the cell wall of fungi and other organisms [31] including dinoflagellates such as *A. ocellatum* [32,33].

A similar affinity is documented for wheat germ agglutinin (WGA). This plant lectin binds to N-acetyl-D-glucosamine and most strongly to its oligomers or polymers as chitin [34,35]. The labelling results obtained for WGA and WGA + rhodamine are consistent with those obtained with CFW. Despite the different emission wavelengths, the usage of these stains contributed to the specific labelling of the cellulose wall of attached trophonts.

In conclusion, CFW, DAPI, PI and WGA provided specific labelling of the cellulose wall, cytoplasmic granules and nuclei. Conversely, the anchoring organs and stomopode were not well stained and were visible only with eosin. Unfortunately, the thickness of H-E stained sections was not adequate for this technology, so it was not possible to appreciate the disposition of rhizoids in the gill cells nor the entire length of the stomopode. In any case, the results collected through this research suggest that both rhizoids and stomopode are composed of proteins, which enable these organs to contract and move thus allowing trophonts to anchoring and nutrients absorption. However, further studies will be necessary to clarify this finding by testing different stains specific for protein marking in order to identify them and better comprehend if they could be antigenic.

One of the aims of this investigation was to clarify and possibly confirm the feeding function of the stomopode through an ultrastructural approach based on the study by Lom and Lawler [12] by means of transmission electron microscopy (TEM). Unfortunately, during the several steps of sample processing, most trophonts detached from the gill epithelium. Furthermore, despite several ultrathin sections produced, no stomopode was observed passing into the cells and only in one trophont were some rhizoids evident. Starch granules, food vacuoles and a large nucleus were the main structures observed in the cytoplasm of trophonts. Due to the limited results obtained we have preferred not to include in this paper the TEM investigations performed.

The approaches applied in this study have allowed a better comprehension of how AO interacts with its host and how the host responds to it. Moreover, it was also possible to better characterize the anatomy and morphology of AO through the use of novel approaches. The knowledge provided by this study lays a foundation for future research aimed at formulating better targeted treatments against AO, thus limiting the dramatic impacts that this protozoan can have on the semi-intensive aquaculture systems worldwide. In fact, the comprehension of what proteins compose rhizoids and stomopode, the organs most in contact with host epithelia, and if these proteins may be antigenic could contribute to the development of a vaccine, thus leading to a more effective and durable protection against the infestation.

## 4. Materials and Methods

Infested tissues of ESB juveniles, obtained from spontaneous outbreaks (4 natural infestations during which 40 fish were sampled) and from experimental infestation challenges (see Section 4.2) were employed for histology (HIS) and immunohistochemistry (IHC). Dedicated paraffin-embedded blocks and/or slides stained following different staining protocols have been analysed by confocal microscopy (CLSM). Tissues of healthy ESB were also collected in order to provide controls for some laboratory tests. For lethal sampling, all fish were euthanized through terminal anaesthesia using tricaine methanesulfonate (MS-222, 400 mg L^−1^) (Sigma-Aldrich, Saint Louis, MO, USA).

### 4.1. Histology (HIS)

Gill and head samples collected from healthy or infested ESB juveniles were fixed either in Bouin’s solution overnight at 4 °C or in 4% buffered formaldehyde and subsequently embedded in paraffin by standard histological protocols, sectioned (4 µm in thickness) and stained with haematoxylin and eosin (H-E), Periodic acid Schiff (PAS)-Alcian blue, Masson and Azan trichrome, Cleveland trichrome, Giemsa and Twort Gram. The specimens were evaluated under light compound microscope (Leica DMLB) and relevant images were captured using a digital camera (LEICA ICC50) with imaging software (LAS EZ V1).

### 4.2. Immunohistochemistry (IHC)

Twenty-six ESB (mean weight 14 g ± 1.7) per tank were bath challenged in triplicate for 2 h with AO under controlled conditions (26–28 °C and 34‰ salinity). Before the infestation, water volume in the tanks destined to the infestation trial was reduced to 50 L. Dinospores, obtained from trophonts collected during natural European seabass infestation, were counted as per Dehority [36], and added to the specific tanks at a final concentration of 3.5 × 10^6^ tank^−1^. As a control group (non-infested), 26 ESB per tank in triplicate were also used. The time-course of infestation was determined by fresh microscopical examination of gills. After 2 h, fish were already infested, and at the second day post infestation (dpi) the trophont burden was discrete with light clinical symptoms. During the infestation, the maximum AO burden was observed at 10–12 dpi. Thereafter, fish started to recover even if positive for amyloodiniosis. Throughout this period, the total mortality was 18%. In the control group, fish were not infested, and no mortality was registered. During the infestation (Days 1, 3, 5, 30) gills of three fish/sampling were collected, processed for histology and 5 µm sections dried overnight on adhesive glass slides (TOMO^®^ Matsunami, St. Ingbert, Germany) and used for the IHC tests according to the protocol reported in Byadgi et al. [17].

To evaluate the tissue host response in terms of antibodies and products recruited to combat AO over the time-course of infestation, polyclonal antibodies specific for European sea bass IgM (1:24,000; University of Trieste, Trieste, Italy) and Inducible Nitric Oxide Synthase (iNOS) (1:250; RB-1605, Thermo Scientific, Waltham, MA, USA) were employed for assessment.

### 4.3. Confocal Laser Scanning Microscopy

Confocal investigations were performed at the Institute of Aquaculture (IoA, University of Stirling, Stirling, UK). All slides were visualized using a Leica SP2 AOBS multi-spectral confocal laser scanning microscope (CLSM) and examined with a 405 nm diode laser and HeNe (543 nm) laser. By xyz mode scanning of samples at various focal planes along the Z-axis a three-dimensional data set was acquired for samples and the morphology of the parasite and interaction with host tissues were evaluated.

For this study Bouin’s fixed ESB AO infested gills, sectioned at 4 µm, were prepared using standard histological protocols and stained with haematoxylin and eosin at UNIUD.

In parallel, some ESB AO infested gill samples, deriving from an experimental infestation, were subjected to different staining and labelling protocols in order to obtain better resolution of the features of interest. Details of the experimental infestation challenge and the staining/labelling protocols applied are reported below.

#### 4.3.1. Experimental Infestation Challenge and Sample Collection

Ten healthy ESB (mean weight 70 g) were subjected to an experimental infestation with *Amyloodinium ocellatum* at UNIUD facilities. The trial was performed in a 300 L tank, half filled with 34‰ seawater, at 27 ± 2 °C. Dinospores were counted as described by Dehority [36] and added to the tank at a final concentration of 4 dinospores mL^−1^. Fish were constantly monitored and at the onset of symptoms (5 dpi) euthanized (MS-222, 400 mg L^−1^; E10521, Sigma-Aldrich). Gill samples were fixed in 4% paraformaldehyde in PBS (pH 7.4) (16005 & P5368, Sigma-Aldrich) overnight at 4 °C, then transferred to 70% ethanol until use. Sample paraffin embedding was performed at IoA. Ten µm sections were cut from the 4% paraformaldehyde-fixed, wax-embedded tissues and mounted on Plus Frost positively charged microscope slides (MSS51012WH, Solmedia, Shrewsbury, UK). Some control gill samples were collected from ESB unexposed to *A. ocellatum* and similarly processed.

#### 4.3.2. DAPI and TRITC-Phalloidin Labelling

Ten µm 4% paraformaldehyde-fixed paraffin-embedded samples were dewaxed twice with xylene (3 min and 2 min) and rehydrated using 100% ethanol for 2 min, followed by immersion in methylated spirit for 1.5 min. Slides were then washed with tap water for 1 min and stained with eosin (5 min). Then, a proportion of the slides was mounted using a 4,6-diamidino-2-phenylindole dihydrochloride (DAPI) mountant medium (H-1200, Vectashield, Vector Laboratories, Burlingame, CA, USA), while the remaining slides were mounted with a TRITC (tetramethylrhodamine)-phalloidin mountant medium (H-1600, Vectashield, Vector laboratories) for nuclear and cytoskeleton staining respectively.

#### 4.3.3. Double Staining with Calcofluor White and Propidium Iodide

A double-staining method combining two fluorescent stains was also tested. Ten µm 4% paraformaldehyde-fixed paraffin-embedded samples were dewaxed and rehydrated as described for the DAPI and TRITC-phalloidin staining. Slides were then washed with tap water for 1 min and stained with eosin for 1 min (since with 5 min immersion in eosin, the background was too strong). Then slides were rinsed for 1 min with distilled water (dH_2_O) and incubated in the dark with 1:25 propidium iodide (PI) (BMS500PI, ThermoFisher Scientific, Waltham, MA, USA) in dH_2_O in a humidified box for 30 min at room temperature (RT) for nuclear counterstain. Then, in the last 5 min of incubation with PI, the specific chitin/cellulose binding fluorochrome calcofluor white (CFW) (F3543, Sigma Aldrich, St. Louis, MO, USA) was employed at a concentration of 1% CFW stock solution. CFW stock solution was prepared as described by Rasconi et al. [31]. After incubation, slides were washed with dH_2_O for 1 min and mounted with an aqueous Antifade Mounting Medium (H-1000, Vectashield, Vector Laboratories). Finally, coverslip edges were sealed with clear nail varnish.

#### 4.3.4. Fluorescent Lectin Labelling

This labelling approach was based on a protocol developed at Institute of Aquaculture. Wheat germ agglutinin (WGA) lectin obtained from fluorescein Lectin Kit I (FLK-2100, Vector Laboratories) and WGA plus rhodamine obtained from Lectin Kit I (RLK-2200, Vector Laboratories) were selected to label the carbohydrates of AO trophonts. Lectins were diluted to a final concentration of 5 µg/mL in the lectin wash buffer (LWB) (50 mM Tris(hydroxymethyl)aminomethane, 150 mM NaCl, 2 mM MgCl_2_ and 1 mM CaCl_2_; pH 7.4, all compounds were purchased from Sigma-Aldrich). Then 5 µm sections fixed in 4% paraformaldehyde and paraffin-embedded were dewaxed into two changes of xylene for 3 min each, followed by rehydration using 100% ethanol (2 min) and 70% ethanol (2 min). After washing in dH_2_O for 1 min, 200 µL of lectin solution was pipetted onto the dewaxed sections and incubated in a dark chamber for 2 h at RT. Thereafter, sections were washed in LWB three times for 5 min each. A negative control was used for each lectin labelling investigation and treated in the same way as test lectin sections, but with the use of LWB only. Slides were washed in LWB (3 × 5 min), then mounted in a DAPI mountant (H-1200, Vectashield, Vector Laboratories), coverslipped and sealed with clear nail varnish. Slides were then incubated in the dark for 30 min to maximise staining.

### 4.4. Animal Ethics

The experimental infestation challenges reported in the present study were carried out in the facilities (fish stabularium, ID 5E7A0) of the Department of Agricultural, Food, Environmental and Animal Sciences (University of Udine, Udine, Italy), as authorized by the Italian Ministry of Health (decree n 14/2018-UT, 12/11/2018). The animal care and protocols adopted adhere to Directive 2010/63/EU of the European Parliament, implemented at a national level by the D.L. n. 26 of 4 March 2014.

## Figures and Tables

**Figure 1 pathogens-11-00062-f001:**
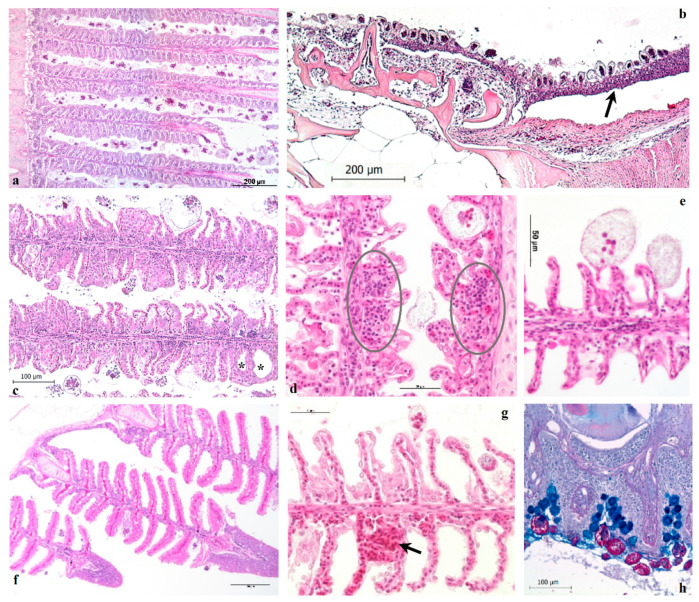
European sea bass. (**a**,**c**) Heavily infested gills with trophont adhesion inducing severe and diffuse epithelial hyperplasia and degeneration with fusion of secondary lamellae in which synechiae (asterisks) are easily visible at lamellar tips. (**b**) Floor of buccal cavity showing anchored trophonts (arrowheads) responsible for the epithelial hyperplasia (arrow) and necrosis. (**d**) Focal lymphocytic infiltrate in gill lamellae (ringed); scale bar = 25 μm. (**e**) High magnification of trophont adhesion apparatus on gill epithelial cells which cause hydropic cell degeneration (also of chloride cells), oedema, and necrosis. (**f**) Some trophonts attached to pseudobranchs, where it is possible to observe moderate oedema and epithelial degenerative processes; scale bar = 100 μm. (**g**) Vascular damage characterized by telangiectasia formation (arrow) and micro haemorrhages; scale bar = 25 μm. (**h**) Pharyngeal tract with attached AO trophonts in which starch granules are stained in purple-magenta and very abundant mucous cells (stained in blue) are visible in the epithelium; PAS—Alcian blue. All images with the exception of figure (**h**) are stained with H-E.

**Figure 2 pathogens-11-00062-f002:**
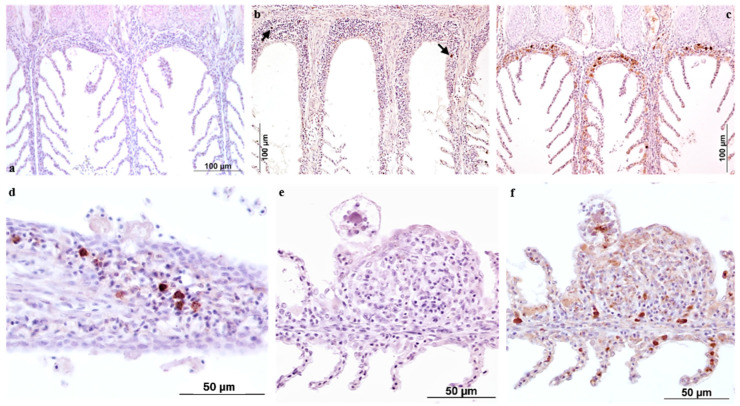
European sea bass. Immunohistochemical staining of AO infected gills in paraffin sections. (**a**,**e**) Omission of the primary antibody. (**b**,**c**) Immunoreactivity to ESB IgM at 5 and 30 days post infestation respectively, plasma cells or IgM bearing macrophages are located in the pockets between primary lamellae; (**d**) magnification of immunoreactivity at the tip of a primary lamellae at 30 days post infestation. (**e**,**f**) iNOS immunoreactivity of cells in a hyperplastic portion of gills at 3 days post infestation; an attached trophont displays positivity for iNOS.

**Figure 3 pathogens-11-00062-f003:**
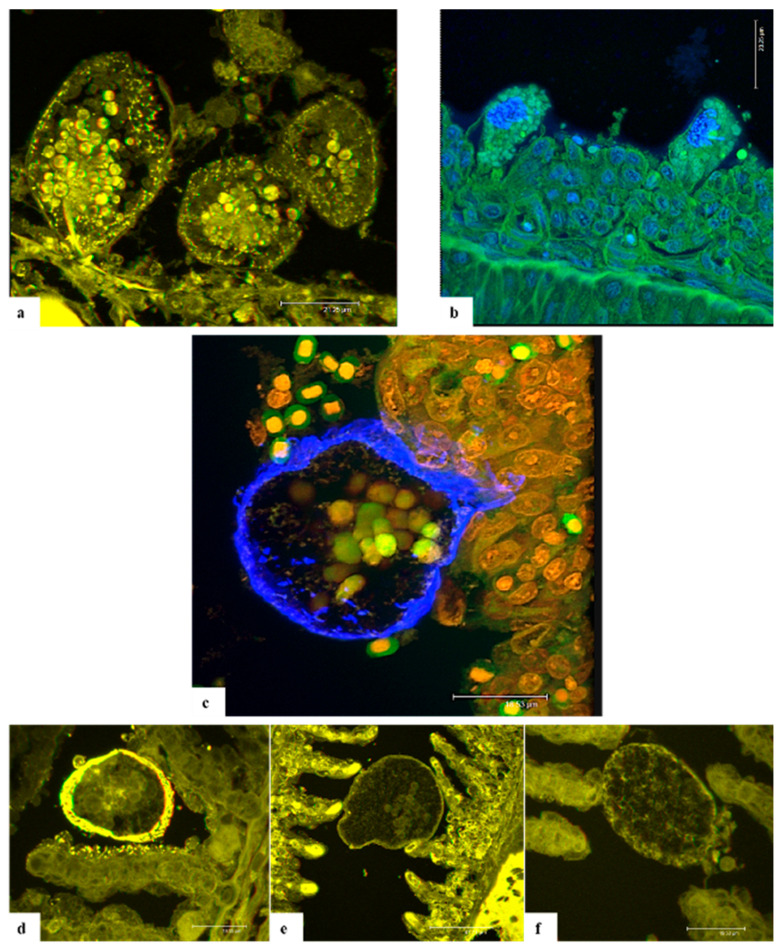
*A. ocellatum* trophonts attached to the gill epithelium of ESB. Images captured with CLSM. (**a**) H&E staining defined the AO cytoplasmic protein granules and the stomopode (arrow). Scale bar = 21.25 μm. (**b**) Eosin and DAPI labelling, AO nucleus and gill cell nucleus light blue fluorescence due to DAPI. Scale bar = 23.25 μm. (**c**) CFW and PI double staining. PI stained nuclei in red; the trophont cellulose wall is blue labelled by CFW; green fluorescence is due to eosin counterstaining. Scale bar = 18.53 μm. (**d**) WGA labelling of AO cellulose wall. Scale bar = 19.55 μm. (**e**) WGA + rhodamine labelling of AO cellulose wall. Scale bar = 47.71 μm. (**f**) Lectin labelling negative control, incubation with dilution buffer only. Scale bar = 19.53 μm. Images (**a**,**d**,**e**,**f**) are 3D anaglyphs.

## Data Availability

Not applicable.

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
