# Peer review of "News Insights into the Host-Parasite Interactions of Amyloodiniosis in European Sea Bass: A Multi-Modal Approach"

_pathogens, 2022, doi:10.3390/pathogens11010062_

Round 1

Reviewer 1 Report

The manuscript provides interesting and practical data on the Amyloodinium ocellatum and amyloodiniosis. Due to no efficacious prophylactic or therapeutic controls for amyloodiniosis, each new information about biology A. ocellatum is useful.

I am of an opinion that the article fits into scope of Pathogens.

Comments:

When the species (parasite, hosts) is first entered, the author/s and the date of the species should be provided, e.g. Dicentrarchus labrax (Linnaeus, 1758).

Line 30: I suggest adding also a kingdom Chromista.

Lines 44, 55-56 etc, etc.: should be [2–6] and [4, 5,7–10] etc.

Line 289: “…by 288 Lom and Lawler (1973)…” – should be “…by Lom and Lawler [11]".

Line 356, etc.: subtitles should be italicized.

Lines 436-515: References list inconsistent with the requirements of the journal, please correct.

Very nice confocal photos !

Author Response

Thanks for the appreciation of our work and the images. We have done what required.

Reviewer 2 Report

Regarding the abstract : 

Line number (LN) 10---add is /LN:11---add and is /LN:13---add bad before impacts/LN:16----add of instead of targeting /LN;19----remove entire/LN:20---add branchial cell hyperplasia as well as cells of oropharyngeal cavity.

LN:20---remove (IHC)/LN:21----plasma cells and macrophages---what about eosinophils that increased with parasitic infestation????

LN:26&27-----add Sea bass to the key words

Other corrections in the attached files should be done 

Author Response

We accepted some of the changes proposed by the auditor, while others were argued by us as not accepted.